# ATTENTION PROJECTION MIXING WITH EXOGENOUS ANCHORS

## ABSTRACT

Cross-layer reuse of early attention projections can improve optimization and data efficiency, but it creates a structural conflict: the first layer must simultaneously act as a stable, reusable anchor for all deeper layers and as an effective computational block. We demonstrate that this tension constrains the performance of internal-anchor designs. We propose ExoFormer, which resolves the conflict by learning *exogenous anchor projections* outside the sequential layer stack. We introduce a unified normalized mixing framework that mixes queries, keys, values, and gate logits using learnable coefficients (exploring coefficient granularities: elementwise, headwise, and scalar), and we show that normalizing anchor sources is key to stable reuse. ExoFormer variants consistently outperform their internal-anchor counterparts, and the dynamic variant yields $\sim 1.5$ downstream accuracy points while matching validation loss using $\sim 1.5\times$ fewer tokens than Gated Attention. We explain this efficacy via an *Offloading Hypothesis*: external anchors preserve essential token identity, allowing layers to specialize exclusively in feature transformation. We release code and models to facilitate future research.

## 1 INTRODUCTION

The Transformer architecture (Vaswani et al., 2017) underpins modern large language models (LLMs) and contextual tasks. Its success relies on the multi-head self-attention mechanism, which enables dynamic, context-dependent interactions across sequences. However, as models scale in depth, ensuring stable training and effective information propagation remains a challenge.

Token information is diluted in deeper layers due to over-smoothing (Shi et al., 2022; Zhou et al., 2021), spurring interest in direct mechanisms for preserving early representations. Existing solutions leave key questions unanswered; ResFormer (Zhou et al., 2025) focuses solely on residualizing *values*, leaving the reuse of queries, keys, and gate logits unexplored.

We introduce a unified framework for cross-layer mixing across all attention pathways, systematically evaluating the contribution of mixing queries ($Q$), keys ($K$), and gate logits ($G$), alongside the established value ($V$) residual, using various coefficient granularities. A key insight is that applying RMSNorm to residual sources before mixing resolves distributional mismatch, enabling stable reuse.

Reusing first-layer projections reveals a tension: the layer must serve as both a stable anchor for deeper layers and a computational block for feature transformation. This dual objective inherently limits effectiveness in both roles.

We introduce **ExoFormer**, which resolves this tension by learning *dedicated exogenous anchor projections* outside the sequential layer stack. We compare this with NuResFormer (**N**ormalized **u**nified), an internal-anchor baseline. Decoupling the two roles proves beneficial: ExoFormer variants outperform NuResFormer counterparts in perplexity while remaining competitive in downstream accuracy, without increasing width or depth. We explain this via an *Offloading Hypothesis*: the external anchor preserves token identity, allowing sequential layers to specialize in refinement.

Our empirical analysis reveals that ExoFormer layers spend approximately two-thirds of their depth in the refinement stage (versus one-third in standard Transformers), with catastrophic failure upon anchor removal, confirming that layers specialize under the assumption of externalized identity preservation.

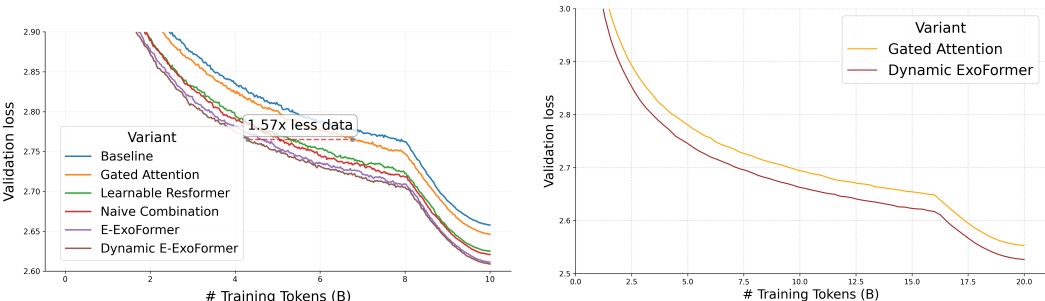

Figure 1: (Left) Validation loss for $\sim 450M$ parameter models. (Right) Validation loss for $\sim 1B$ parameter models.

## 2 METHODOLOGY

We utilize standard Multi-Head Attention (MHA) enhanced with rotary position embeddings (RoPE) and query/key normalization (QKNorm). We incorporate element-wise multiplicative gating (Qiu et al., 2025), modulating the attention output before the final projection. Ablations indicate that post-norm substantially degrades Gated Attention performance, so we default to pre-norm. A comprehensive overview of the methodology can be found in the appendix.

### 2.1 UNIFIED MIXING FRAMEWORK

We propose a framework to enrich attention pathways by mixing current layer projections $S_n$ with persistent "anchor" projections $S_{\text{anc}}$ (where $S \in \{Q, K, V, G\}$). The anchor is normalized and combined with the current projection using learnable coefficients $\lambda$. To stabilize the mixture, we apply RMS normalization to the anchor source before scaling:

$$\widehat{S}_n = \lambda_{n,1}^S \odot \text{RMSNorm}(S_{\text{anc}}) + \lambda_{n,2}^S \odot S_n, \qquad \forall S \in \{Q, K, V, G\}, \tag{1}$$

where $\lambda_{n,1}^S, \lambda_{n,2}^S$ are coefficient tensors of the chosen granularity (scalar, headwise, or elementwise). We provide a detailed ablation of normalization strategies and coefficient granularities in Appendix C.1 and C.2.

**Anchor Instantiations.** We explore two anchor definitions: in NuResFormer, anchors are the projections from the first attention layer $(Q_1, K_1, V_1, G_1)$. Conversely, in ExoFormer, anchors are produced by a dedicated external module on the input embeddings $H_0$.

**Motivation for ExoFormer.** Cross-layer residual reuse makes early information available at every depth. This is powerful, but it implicitly forces the first layer to satisfy two pressures:

1. **Reusable anchor:** produce a broadly useful reference representation that remains valuable throughout depth.
2. **Progressive computation:** produce features that are easy for downstream layers to transform into increasingly task-relevant abstractions.

While these roles appear misaligned (universal anchors favor invariance, while progressive computation necessitates change), they can theoretically coexist because it is an *optional pathway* modulated by learned mixing coefficients $(\lambda_{n,1}, \lambda_{n,2})$. However, this structural constraint places pressure on the first layer to make compromises, inherently limiting its effectiveness in both roles. This tension is empirically supported by the analysis in Figure 5, showing that NuResFormer's first layer adopts a permissive gating policy compared to standard baselines, indicating a compromise on selectivity in favor of serving as a stable anchor.

**Dynamic Mixing (DM)** Building on the unified framework, we implement a Dynamic Mixing module. We compute context-dependent scaling factors $\gamma$ using a small MLP operating on the layer input $H_{n-1}$. The exact MLP architecture and mixing equations are detailed in the appendix.

## 3 EXPERIMENTS

### 3.1 EXPERIMENTAL SETUP

All models use a modern pre-normalized Transformer architecture with SwiGLU activations (Shazeer, 2020), QKNorm (Henry et al., 2020), and rotary position embeddings (Su et al., 2024) trained on FineWeb-Edu (Penedo et al., 2024) (10B tokens for $\sim 450M$ models; 20B for $\sim 1B$ models. Further details and evaluation setup can be found in the appendix.

Table 1: Parameter counts are 453M (with Gated Attention), 454M (without), and 457M (external anchor). Naïve Combination refers to the unmodified addition of Gated Attention and ResFormer. "No Norm" models omit anchor normalization. Prefixes E, H, and S denote elementwise, headwise, and scalar mixing granularity.

| Model | ARC-c | ARC-e | Hella. | OBQA | PIQA | Wino. | Avg. Acc | PPL |
|---|---|---|---|---|---|---|---|---|
| Base Transformer | 30.97 | 63.38 | 42.02 | 33.60 | 67.30 | 51.54 | 48.14 | 14.79 |
| Gated Attention | 30.89 | 64.69 | 42.73 | 34.00 | 67.14 | 53.35 | 48.80 | 14.64 |
| ResFormer (Value Residual) | 33.62 | 64.86 | 43.49 | 34.40 | **68.88** | 52.64 | 49.65 | 14.32 |
| Naïve Combination | 32.94 | 64.52 | 43.47 | 32.60 | 68.34 | 52.64 | 49.09 | 14.25 |
| E-NuResFormer | **33.70** | 65.07 | 44.23 | 33.60 | 68.34 | 53.12 | 49.68 | 14.15 |
| E-ExoFormer (No Norm) | 32.08 | 63.76 | 43.49 | 34.40 | 68.77 | **55.64** | 49.69 | 14.30 |
| E-ExoFormer | 32.42 | 64.65 | 44.28 | **36.40** | 67.74 | 53.59 | 49.85 | 14.13 |
| **Dynamic E-ExoFormer** | 33.36 | **65.87** | **44.54** | 34.40 | 68.28 | 55.17 | **50.27** | **14.09** |

### 3.2 MIX-COMPRESS-REFINE THEORY, GATED ATTENTION, AND RESFORMER

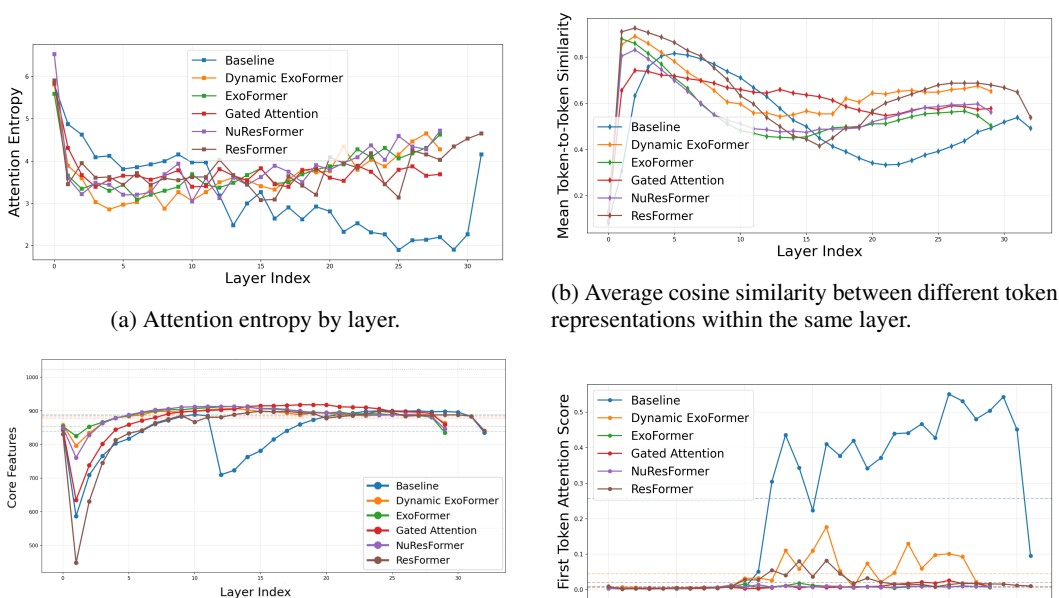

(a) Attention entropy by layer.

(b) Average cosine similarity between different token representations within the same layer.

(c) PCA Core Features by layer to explain 99% variance.

(d) First-token attention (attention sink) by layer.

Figure 2: Attention-pattern and representation analysis across model variants $\sim 450M$. *Elementwise* is used unless stated otherwise. Some graphs include input embeddings for comparison.

As shown in Figure 2, the baseline Transformer's behavior aligns with the Mix-Compress-Refine theory proposed by Queipo-de-Llano et al. (2025): (1) it begins with a high attention entropy phase for broad, contextual integration of token information, (2) transitions into a compression valley

marked by dominant attention sinks and a drastic drop in core features that halt mixing and reduce representational dimensionality to filter out useless information, (3) concludes with a sudden rise in attention entropy as sinks dissipate, enabling the refined, token-specific processing necessary for generation.

While these three stages emerge naturally in standard Transformers, they are not optimized for computational efficiency. The model expends computational capacity collapsing contextual information, only to subsequently reconstruct it, indicating inefficiency in the standard architecture (Figure 2c).

**Gated Attention and residual mixing improve performance by targeting stage 2.** We propose that the performance gains from Gated Attention and ResFormer may partly stem from their effect on the model's second, compression stage.

The gating mechanism addresses the model's need to filter irrelevant context (core function of stage 2). By selectively modulating information flow at every layer, gated attention provides a form of distributed filtering, reducing the need for a sharp, dedicated compression phase. This interpretation is supported by the absence of a clear compression valley and a drastic reduction in attention sink magnitude as shown in Figure 2.

Similarly, in models employing residual mixing, the residual pathways may provide an implicit, learned alternative to abrupt sink-based filtering. The learned blending reduces the model's reliance on extreme, attention-sink-driven compression to isolate useful signals. This is supported by the milder compression valley observed in such models and their notable reduction in attention sink magnitude (Figure 2)

### 3.3 THE OFFLOADING HYPOTHESIS: SPECIALIZATION VIA EXOGENOUS ANCHORS

The architectural decoupling in ExoFormer enables an interesting functional specialization, which we formalize as the Offloading Hypothesis. By providing a dedicated, high-fidelity source of token identity, the exogenous anchor allows sequential layers to offload the preservation of static features and specialize almost exclusively in the final "refinement" stage (Stage 3).

This specialization is evident in token-similarity trajectories (Figure 2b), where a local minimum marks the onset of Stage 3. Crucially, ExoFormer variants and NuResFormer spend approximately two-thirds of their layers in this refinement stage, the highest proportion of any model, compared to one-third for the baseline.

In a standard architecture, the residual stream must concurrently carry two conflicting types of information: (1) distinct, static features that preserve token identity ("What token am I?"), and (2) transformed, task-relevant features that evolve through the layers to support next-token prediction. Over-smoothing in standard models catastrophically loses the first type, crippling the model's ability to route information effectively. ExoFormer circumvents this tension by introducing a persistent external anchor that is reinjected at every layer, guaranteeing access to high-fidelity token identity.

When we remove the exogenous anchor during inference, the model suffers a catastrophic failure: core features plummet drastically and reach 321 in the final layer, while token-to-token similarity peaks at 93%. This dramatic collapse demonstrates that the sequential layers, when deprived of their dedicated identity source, lose most discriminative power. The layers were optimized under the specific assumption that identity preservation was offloaded to the anchor. When that anchor is removed, they have no mechanism to maintain token distinctiveness.

## 4 CONCLUSION

We introduced ExoFormer, a novel Transformer architecture that decouples token identity preservation from computational refinement by learning dedicated exogenous anchor projections. Experiments demonstrate that this decoupling consistently improves performance, with the dynamic variant achieving significant gains in downstream accuracy and data efficiency compared to standard gated attention baselines. We also partially explain the empirical efficacy of Gated Attention and ResFormer. Our results validate the Offloading Hypothesis, suggesting that externalizing identity preservation enables sequential layers to specialize exclusively in high-level feature transformation, providing an efficient architectural path for enhancing large language models.

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

## A    EXTENDED METHODOLOGY

This section provides comprehensive mathematical details and definitions that support the methodology introduced in Section 2 of the main text.

### A.1    PRELIMINARIES AND NOTATION

For the $n$-th Transformer layer, let $H_{n-1} \in \mathbb{R}^{T \times d_{\text{model}}}$ be the input hidden states (after prenormalization), where $T$ is the sequence length and $d_{\text{model}}$ is the model width. We use $h$ attention heads and per-head dimension $d_k$ such that $d_{\text{model}} = h\, d_k$. We denote the projected queries, keys, values, and gate logits as $Q_n, K_n, V_n, G_n$. Let $\text{RMSNorm}(\cdot)$ denote per-token RMS normalization applied independently to each attention head (with learnable gain).

We consider three coefficient granularities for $\lambda$: Scalar (S) (a single coefficient shared across all channels), Headwise (H) (one coefficient per head, broadcast across $d_k$), and Elementwise (E) (one coefficient per channel). Elementwise and headwise granularities grant fine-grained control over reuse mechanisms. We further analyze how they impact training and downstream performance.

## A.2 MULTI-HEAD ATTENTION DETAILS

We describe attention as a sequence of stages, with two standard enhancements integrated implicitly: rotary position embeddings (RoPE) (Su et al., 2024) and query/key normalization (QKNorm) (Henry et al., 2020). For simplicity, the multi-head mechanism is presented using unified tensors with an implicit head dimension.

**Stage 1: QKV Linear Projections.** Given $H_{n-1} \in \mathbb{R}^{T \times d_{\text{model}}}$, we compute the projected tensors:

$$Q_n = H_{n-1}W_n^Q, \quad K_n = H_{n-1}W_n^K, \quad V_n = H_{n-1}W_n^V, \tag{2}$$

where $W_n^Q, W_n^K, W_n^V \in \mathbb{R}^{d_{\text{model}} \times d_{\text{model}}}$. The resulting $Q_n, K_n, V_n$ are structured to contain $h$ heads implicitly.

**Stage 2: Scaled Dot-Product Attention (SDPA).** Attention is computed per head, which is represented here as a single operation:

$$A_n = \text{softmax}\left(\frac{Q_n K_n^\top}{\sqrt{d_k}}\right) \in \mathbb{R}^{T \times T},$$
$$U_n = A_n V_n \in \mathbb{R}^{T \times d_{\text{model}}}, \tag{3}$$

where the operations encompass the independent computations across $h$ heads.

**Stage 3: Final Output Projection.** The output of the attention computation is projected:

$$O_n = U_n W_n^O, \qquad W_n^O \in \mathbb{R}^{d_{\text{model}} \times d_{\text{model}}}. \tag{4}$$

In NuResFormer/ExoFormer, the above stages use mixed tensors $\widehat{Q}_n, \widehat{K}_n, \widehat{V}_n$ as defined in Eq. equation 1 in the main text.

## A.3 UNIFIED MIXING FORMULATION

The core mixing mechanism is defined in Eq. equation 1 of the main text. Here we detail the specific instantiations of the anchors used in our experiments.

**Anchor Instantiations.** We explore two anchor definitions. In NuResFormer, anchors are the projections from the first attention layer $(Q_1, K_1, V_1, G_1)$. Conversely, in ExoFormer, anchors are produced by a dedicated external module on the input embeddings:

$$Q_{\text{anc}} = H_0 W_{\text{anc}}^Q, \quad K_{\text{anc}} = H_0 W_{\text{anc}}^K, \quad V_{\text{anc}} = H_0 W_{\text{anc}}^V, \quad G_{\text{anc}} = H_0 W_{\text{anc}}^G, \tag{5}$$

where $W_{\text{anc}}^Q, W_{\text{anc}}^K, W_{\text{anc}}^V, W_{\text{anc}}^G \in \mathbb{R}^{d_{\text{model}} \times d_{\text{model}}}$ are independent learnable weight matrices.

## A.4 DYNAMIC MIXING (DM) MODULE DETAILS

Building upon the unified formulation, we introduce a dynamic variant where the learnable parameters are modulated by context-dependent scaling factors computed from the layer input $H_{n-1}$ using a small MLP.

**Dynamic Coefficient Generation.** For each layer $n$, we compute modulation scalars from its input $H_{n-1}$ (pre-normalized) using a two-layer MLP with GELU activation and sigmoid output:

$$\mathcal{DM}_n(H_{n-1}) = \sigma\left(\text{GELU}(H_{n-1}W_{n,1}^{\text{DM}})W_{n,2}^{\text{DM}} + b_n^{\text{DM}}\right) \tag{6}$$

The trainable parameters for the Dynamic Mixing module at layer $n$ are:

$$\theta_n^{\text{DM}} = \left\{W_{n,1}^{\text{DM}} \in \mathbb{R}^{d_{\text{model}} \times 16}, \ W_{n,2}^{\text{DM}} \in \mathbb{R}^{16 \times 8}, \ b_n^{\text{DM}} \in \mathbb{R}^8\right\}$$

The output dimension of this module is 8, corresponding to the dynamic scaling factors:

$$\{\gamma_{n,1}^Q, \gamma_{n,2}^Q, \gamma_{n,1}^K, \gamma_{n,2}^K, \gamma_{n,1}^V, \gamma_{n,2}^V, \gamma_{n,1}^G, \gamma_{n,2}^G\}$$

The output layer weights $W_{n,2}^{\mathrm{DM}}$ and bias $b_n^{\mathrm{DM}}$ are zero-initialized, ensuring that initial sigmoid outputs are 0.5. Consequently, the base $\lambda$ parameters must be initialized at 1.0 to achieve effective identity mixing at initialization.

**Modulated Mixing.** For each component we compute using the dynamic scaling factors:

$$\widehat{S}_n = (\lambda_{n,1}^S \gamma_{n,1}^S) \odot \mathrm{RMSNorm}(S_{\mathrm{anc}}) + (\lambda_{n,2}^S \gamma_{n,2}^S) \odot S_n, \qquad \forall S \in \{Q, K, V, G\}, \qquad (7)$$

where $\gamma_{n,i}^S$ are broadcast appropriately based on the residual granularity (elementwise, headwise, or scalar), and $\lambda_{n,i}^S$ are the learnable base parameters.

# B  RELATED WORK

**Value Residuals and Gated Attention.** Zhou et al. (2025) introduced *Value Residual Learning* (ResFormer), which adds a residual connection from the first layer's value projection ($V_1$) to the value projections of all subsequent layers. This method was shown to greatly improve model performance and data efficiency. However, preliminary attempts to residualize queries and keys were found to be unstable (Zhou et al., 2025). Concurrently, gated attention mechanisms have been explored to introduce dynamic, input-dependent modulation to the attention output (Qiu et al., 2025). Our work generalizes residual learning to all attention pathways and stabilizes it via normalized mixing, bridging the gap between residual connections and gated attention.

**Cross-layer communication and residual mixing.** Recent work has sought to improve cross-layer information flow in Transformers beyond simple residual connections. Zhu et al. (2025) proposed Hyper-Connections that expand residual stream width. More recently, Xie et al. (2026) introduced Manifold-Constrained Hyper-Connections (mHC), restoring an identity-like signal-preservation property to the hyper-connected architecture.

Most closely related to our work is **MUDDFormer** (Xiao et al., 2025), which proposes Multi-way Dynamic Dense (MUDD) connections. MUDDFormer decouples the input to each Transformer block into four streams and dynamically aggregates outputs from *all* preceding layers using context-specific weights generated by a small MLP.

While MUDDFormer primarily addresses the *input* to the attention projections, our unified mixing framework focuses on *mixing* the attention projections themselves. Our approach can be seen as a form of blending *after* projection, whereas MUDDFormer enriches the input *before* projection.

# C  FULL RESULTS AND ANALYSIS

**Full Training Details** All models use a modern pre-normalized Transformer architecture with SwiGLU activations (Shazeer, 2020), QKNorm (Henry et al., 2020), and rotary position embeddings (Su et al., 2024). We follow prior work by initializing projection and classification layers to zero (Yang et al., 2022), removing bias terms (except for the dynamic mixing module) (Chowdhery et al., 2023), applying z-loss regularization (de Brébisson & Vincent, 2016), and disabling dropout. Training is performed on the FineWeb-Edu dataset (Penedo et al., 2024). Specifically, the $\sim 450$M parameter models are trained on 10B tokens, while the $\sim 1$B parameter models are trained on 20B tokens.

We optimize matrix parameters with Muon (Polar Express variant (Amsel et al., 2025)), applying a cautious weight decay of 0.1 (Chen et al., 2025), while 1D parameters are trained using AdamW without weight decay. We selected Muon as a state-of-the-art optimizer for large-scale LLM training (Liu et al., 2025), though we anticipate that the observed improvements should be optimizer-agnostic. Gradient norms are clipped to 1.0, and all models follow the same optimization setup, to ensure fair comparison across architectures. Training uses a global batch size of 262,144 tokens and a sequence length of 2,048.

Full hyperparameters are provided in the Appendix. All experiments are conducted on a single NVIDIA H100 80GB GPU using native BF16 precision, with FlashAttention (Dao et al., 2022) enabled.

**Evaluation Details** We evaluate each benchmark example using a 5-shot prompt. To reduce length-related bias, we report length-normalized accuracy whenever possible. Perplexity is measured on the FineWeb-Edu validation set containing 100 million tokens.

We report results on 6 multiple-choice benchmarks: ARC_CHALLENGE, ARC_EASY (Clark et al., 2018), HELLASWAG (Zellers et al., 2019), OPENBOOKQA (Mihaylov et al., 2018), PIQA (Bisk et al., 2020), and WINOGRANDE (Sakaguchi et al., 2020).

Table 2: Full performance comparison of model variants on 6 multiple-choice downstream tasks. Parameter counts are 453M (with Gated Attention), 454M (without), and 457M (external anchor). Naïve Combination refers to the unmodified addition of Gated Attention and ResFormer. Prefixes E, H, and S denote elementwise, headwise, and scalar mixing granularity. "Only Q/K Norms" applies RMSNorm solely to anchor queries/keys, while "No Norm" omits anchor normalization.

| Model | ARC-c | ARC-e | Hella. | OBQA | PIQA | Wino. | Avg. Acc | PPL |
|---|---|---|---|---|---|---|---|---|
| **Baselines** | | | | | | | | |
| Base Transformer | 30.97 | 63.38 | 42.02 | 33.60 | 67.30 | 51.54 | 48.14 | 14.79 |
| Gated Attention | 30.89 | 64.69 | 42.73 | 34.00 | 67.14 | 53.35 | 48.80 | 14.64 |
| ResFormer (Value Residual) | 33.62 | 64.86 | 43.49 | 34.40 | **68.88** | 52.64 | 49.65 | 14.32 |
| Naïve Combination | 32.94 | 64.52 | 43.47 | 32.60 | 68.34 | 52.64 | 49.09 | 14.25 |
| **Internal Anchor** | | | | | | | | |
| E-NuResFormer (Only Q/K Norms) | 31.74 | 64.27 | 43.83 | 33.00 | 68.23 | 52.41 | 48.91 | 14.21 |
| H-NuResFormer (Only Q/K Norms) | **34.73** | 64.06 | 44.45 | 34.20 | 67.57 | 53.04 | 49.68 | 14.21 |
| S-NuResFormer (Only Q/K Norms) | 32.85 | **66.08** | 43.87 | 34.20 | 68.28 | 53.67 | 49.83 | 14.22 |
| E-NuResFormer (No Norm, {V,G}) | 32.42 | 64.94 | 43.80 | 35.00 | 67.52 | 51.54 | 49.20 | 14.24 |
| H-NuResFormer (No Norm, {V,G}) | 31.23 | 64.18 | 43.50 | 34.00 | 67.68 | 54.06 | 49.11 | 14.22 |
| S-NuResFormer (No Norm, {V,G}) | 31.83 | 64.44 | 43.66 | 34.80 | 68.44 | 52.49 | 49.28 | 14.24 |
| E-NuResFormer | 33.70 | 65.07 | 44.23 | 33.60 | 68.34 | 53.12 | 49.68 | 14.15 |
| H-NuResFormer | 33.79 | 65.11 | 44.09 | 33.40 | 67.41 | 52.72 | 49.42 | 14.17 |
| S-NuResFormer | 32.42 | 64.02 | 43.93 | 33.20 | 67.90 | 53.20 | 49.11 | 14.17 |
| **External Anchor** | | | | | | | | |
| E-ExoFormer (No Norm) | 32.08 | 63.76 | 43.49 | 34.40 | 68.77 | **55.64** | 49.69 | 14.30 |
| **Dynamic E-ExoFormer** | 33.36 | 65.87 | **44.54** | 34.40 | 68.28 | 55.17 | **50.27** | **14.09** |
| E-ExoFormer | 32.42 | 64.65 | 44.28 | **36.40** | 67.74 | 53.59 | 49.85 | 14.13 |
| H-ExoFormer | 32.17 | 65.87 | 44.00 | 32.40 | 68.23 | 52.72 | 49.23 | 14.14 |
| S-ExoFormer | 32.17 | 65.66 | 44.33 | 33.60 | 68.06 | 51.14 | 49.16 | 14.15 |

## C.1 The Role of Anchor Normalization

RMSNorm serves two roles: first, acting as a scale-dependent rational function, it introduces a non-linear transformation into the otherwise linear residual pathway; and second, as a mild isotropization operator, RMSNorm projects representations onto the unit sphere while preserving directional information and removing scale differences. The unit vector retains the relative alignment between dimensions, which encodes semantic and syntactic features.

The necessity of applying RMSNorm to anchor sources is empirically supported by an analysis of the learned mixing coefficients in unnormalized models. For instance, in the ExoFormer variant without residual normalization, the proportion of near-zero coefficients (below 0.001) for $\lambda_1$ (the

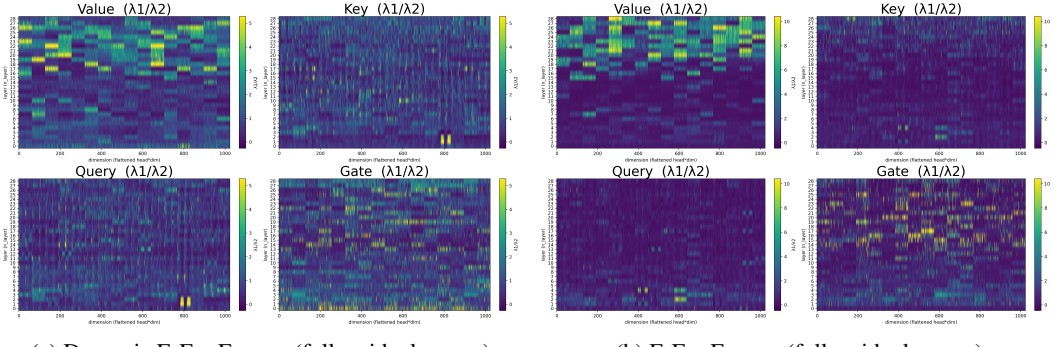

(a) Dynamic E-ExoFormer (full residual norms)          (b) E-ExoFormer (full residual norms)

Figure 3: Heatmaps showing the learned mixing coefficient ratio $\lambda_{n,1}/\lambda_{n,2}$ for each residualized component $\{Q, K, V, G\}$ across layers (y-axis) and channels/heads (x-axis) for model variants $\sim 450M$. This ratio quantifies the model's reliance on the anchor relative to the current layer's projection; a higher value indicates stronger reuse of the early signal.

strength of anchor signal) is approximately triple that of the normalized variant. This suppression of the anchor pathway suggests the model is actively compensating for distributional mismatch.

We hypothesize that the naïve combination underperforms for the above reason. $\sigma(G)$ must compensate for distributional mismatch rather than filtering, resulting in instability during training (Figure 1) and worse downstream accuracy than ResFormer in isolation.

## C.2   EXTENDING MIXING TO Q, K, AND G PATHWAYS

**The Instability of Unnormalized Q/K Residuals and the Stabilizing Role of QKNorm.**   Consistent with prior observations (Zhou et al., 2025), we find a clear hierarchy of stability. Models attempting to use unnormalized Q/K residuals without QKNorm proved difficult to optimize, exhibiting divergent loss in preliminary runs. Critically, adding QKNorm alone stabilizes training and recovers baseline performance. We hypothesize this is because QKNorm compresses the scale of $Q$ and $K$, mitigating the distributional mismatch that arises when injecting early-layer routing signals into deeper layers. Furthermore, adding explicit RMSNorm to the Q/K anchor sources on top of QKNorm yields the best results, enabling positive reuse.

**Gate Logit Mixing Is Inherently More Stable.**   In contrast to $Q$ and $K$, residual connections for the gate logits $G$ were straightforward to incorporate even without normalization. We hypothesize that gate logits, which are compressed by the sigmoid function, are naturally less sensitive to distributional mismatch.

**Granularity in NuResFormer.**   Among NuResFormer configurations, scalar mixing achieves the highest average downstream accuracy (49.83%) despite having the fewest parameters. Headwise mixing performs nearly as well (49.68% to 49.42%), indicating that allocating one degree of freedom per attention head captures a significant portion of the beneficial structure. Elementwise mixing yields the best language modeling perplexity (14.15 under full residual norms) but slightly lower downstream accuracy than its scalar counterpart. This suggests increased parametric freedom can improve in-distribution loss but may not generalize as well.

**Granularity in ExoFormer.**   ExoFormer exhibits a notably different profile. Here, elementwise mixing (E-ExoFormer) achieves the overall best performance, attaining the highest average accuracy (49.85%) and the lowest validation perplexity (14.13) of any static model. This reversal indicates that the effectiveness of coefficient granularity is architecture-dependent. We hypothesize that in this cleaner setting (due to decoupling), the optimizer can effectively exploit fine-grained mixing to orchestrate precise reuse policies without compromising generalization.

**Emergent Head Structure in Elementwise Mixing.**   Visualizing the learned elementwise coefficients reveals emergent patterns specific to each attention component (Figure 3). For the value

pathway ($V$), the heatmaps exhibit *sharp boundaries that align precisely with head blocks*. This structure emerges despite the optimizer having the freedom to set each channel independently, aligning with previous research on heads as specialized submodules for routing information (Voita et al., 2019).

Conversely, queries ($Q$), keys ($K$), and gate logits ($G$) exhibit *finer-grained, intra-head structure*, such as alternating bands of high and low reuse ("striping"), suggesting a form of sub-head specialization. Notably, the positions of these high/low bands in $Q$ often align with those in $K$.

**Flexibility Enabled by Dynamic Mixing.** As shown in Figure 3a, $\lambda_{n,1}/\lambda_{n,2}$ for queries, keys, and gate logits are substantially more uniformly distributed across channels and layers in the dynamic variant. Rather than converging to a fixed, layer-specific reuse policy, the model can modulate the strength of the anchor pathway for each component in real time based on the input sequence, enabling more freedom in expression in the mixing coefficients themselves.

## D    ANCHOR RELIANCE AND HIDDEN STATE SIMILARITY

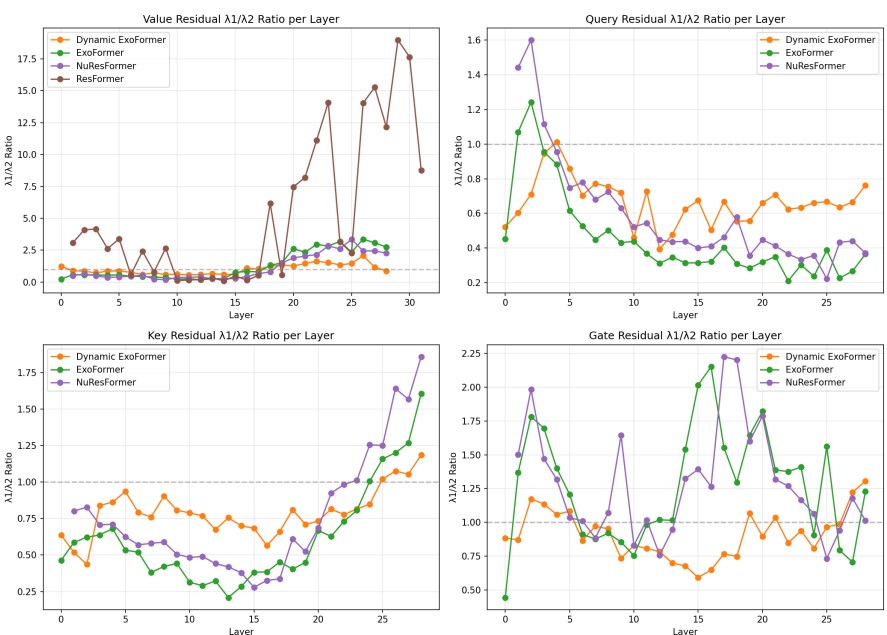

Figure 4: The ratio $\lambda_{n,1}/\lambda_{n,2}$ plotted for each component ($Q$, $K$, $V$, $G$) across layers for models using *elementwise* mixing for model variants $\sim 450M$. Values greater than 1 indicate stronger reliance on the anchor signal, while values less than 1 indicate preference for current-layer projections.

## E    GATE ACTIVATION PROFILES AND FIRST-LAYER SELECTIVITY

NuResFormer's first layer exhibits **a high mean gate activation** (approximately 0.4-0.5), indicating its gating mechanism is not highly suppressive, allowing roughly half of the attention output to pass through. This contrasts sharply with standalone Gated Attention, where the first-layer mean activation is significantly lower (approximately 0.2). Following this initial peak, gate activations fall rapidly in intermediate layers before rising steadily again in deeper layers.

This pattern provides direct empirical support for the architectural tension hypothesized. When the first layer also serves as the residual anchor (as in NuResFormer), it faces conflicting objectives: its gate logits $G_1$ must perform effective, context-dependent selection for the first layer's own computation while also producing a reusable anchor signal $G_{\text{anc}}$ for all subsequent layers. The high first-layer gate activation suggests a resolution: the layer adopts a permissive gating policy to ensure the anchor

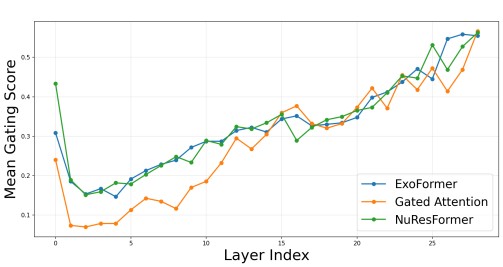 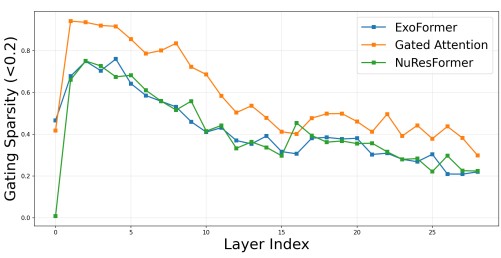

(a) Mean gating score ($\sigma(G)$) per layer. Higher values indicate less suppression of attention output.

(b) Gating sparsity per layer (% activations $< 0.2$). Higher sparsity indicates more selective, suppressive gating.

Figure 5: Analysis of gating behavior across model variants $\sim 450M$.

gate logits retain broad, generally useful information, sacrificing some first-layer selectivity in the process.

ExoFormer exhibits an attenuated version of the same profile (Figure 5a); its first-layer activation is elevated compared to standalone gating but lower than NuResFormer's. As shown in Figure 5b, offloading the anchor role allows ExoFormer to partially restore the first layer's capacity for selective gating.

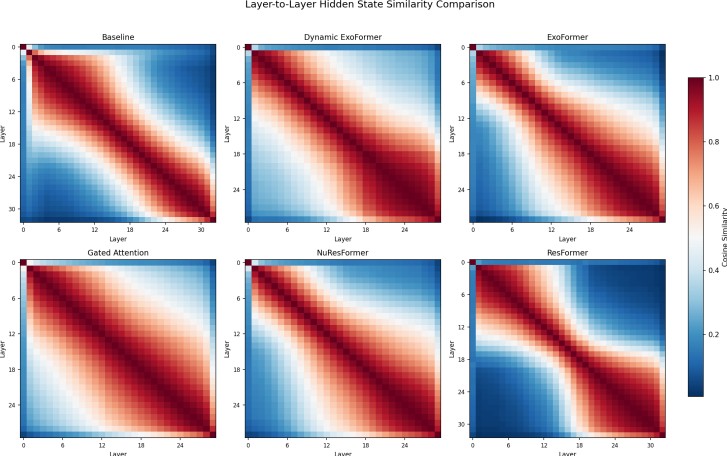

Figure 6: Pairwise cosine similarity of hidden states across transformer layers for models using *elementwise* mixing $\sim 450M$. Brighter colors indicate higher similarity.

## F    COMPLEXITY ANALYSIS

We present a complexity analysis of ExoFormer variants. Let $L$ be the number of Transformer layers and $d$ the model dimension ($d_{\text{model}}$). We omit RMSNorm because it is negligible in terms of parameters and computation.

### F.1    PARAMETER OVERHEAD

We analyze the parameter overhead of ExoFormer variants relative to a baseline Transformer with Gated Attention. Throughout, we ignore input and output embedding parameters, as they are shared across all models.

**Baseline Parameters:**    The baseline Transformer with Gated Attention has parameters per layer for the attention projections (queries, keys, values, gate logits and output) and the two-layer SwiGLU

feed-forward network. The attention and output projections require $5d^2$ parameters (five $d \times d$ matrices), and the FFN requires $6d^2$ parameters (assuming expansion factor 4). Thus, ignoring the input and output embedding, the total parameters for $L$ layers are:

$$P_{\text{base}} = L \cdot \left(5d^2 + 6d^2\right) = 11Ld^2. \tag{8}$$

**Exogenous Anchor Parameters:**  ExoFormer introduces dedicated projection matrices for the exogenous anchor: four $d \times d$ matrices, one for each attention component (Q, K, V, G):

$$\Delta P_{\text{anchor}} = 4d^2. \tag{9}$$

**Static Mixing Parameters:**  For static mixing with elementwise granularity, each layer learns two mixing coefficients (one for anchor, one for current projection) for each of the four components. This amounts to $8d$ parameters per layer:

$$\Delta P_{\text{static}} = L \cdot 8d. \tag{10}$$

**Dynamic Mixing Parameters:**  The Dynamic Mixing (DM) module adds a small MLP per layer. With $d_{\text{DM}} = 16$ and $d_{\text{out}} = 8$, the dominant term is from the first weight matrix ($d \cdot d_{\text{DM}}$). The second weight matrix and biases (totaling 128 parameters) are negligible for typical $d$:

$$\Delta P_{\text{DM}} \approx L \cdot (d \cdot d_{\text{DM}}). \tag{11}$$

**Total Parameter Overhead:**  Static ExoFormer adds:

$$\Delta P_{\text{static ExoFormer}} = 4d^2 + 8Ld. \tag{12}$$

Dynamic ExoFormer additionally includes the DM module:

$$\Delta P_{\text{dynamic ExoFormer}} = 4d^2 + 8Ld + Ld \cdot d_{\text{DM}}. \tag{13}$$

**Parameter Ratio:**  The extra parameter ratio relative to baseline simplifies to:

$$R_P^{(\text{static})} = \frac{4d^2 + 8Ld}{11Ld^2} = \frac{4}{11L} + \frac{8}{11d}, \tag{14}$$

$$R_P^{(\text{dynamic})} = \frac{4d^2 + 8Ld + Ld \cdot d_{\text{DM}}}{11Ld^2} = \frac{4}{11L} + \frac{8 + d_{\text{DM}}}{11d}. \tag{15}$$

For values ($L = 32, d = 1024, d_{\text{DM}} = 16$), this corresponds to approximately 1.2% overhead for static ExoFormer and 1.3% for dynamic ExoFormer, a modest increase given the observed performance gains.

### F.2   COMPUTATIONAL OVERHEAD (FLOPS)

We estimate the floating-point operations (FLOPs) per token during the forward pass, considering matrix-vector multiplications as the dominant factor ($2d^2$ FLOPs per $d \times d$ matrix). We include the cost of elementwise operations for mixing, although they are computationally smaller ($O(d)$) compared to projections ($O(d^2)$).

**Baseline FLOPs:**  The baseline usage corresponds to the parameters utilized at every layer. With $P_{\text{base}} = 11Ld^2$ (excluding embeddings), the computational cost per token is:

$$C_{\text{base}} \approx 2 \cdot P_{\text{base}} = 22Ld^2. \tag{16}$$

**Exogenous Anchor FLOPs:**  The exogenous anchor projections are computed only once per token using the input embeddings, regardless of the network depth. For the four projection matrices ($W_{\text{anc}}^Q, W_{\text{anc}}^K, W_{\text{anc}}^V, W_{\text{anc}}^G$), the cost is:

$$\Delta C_{\text{anchor}} \approx 2 \cdot (4d^2) = 8d^2. \tag{17}$$

Crucially, this cost is constant and does not scale with the number of layers $L$.

**Dynamic Mixing FLOPs:** The Dynamic Mixing module operates at every layer. The computational cost includes the projection of the MLP ($d \to d_{\text{DM}}$) and the elementwise mixing operations. The mixing involves two multiplications (coefficients $\times$ anchor/current projections) and one addition per element for the four components (Q, K, V, G), totaling $12d$ FLOPs per layer:

$$\Delta C_{\text{DM}} \approx L \cdot (2 \cdot d \cdot d_{\text{DM}} + 12d). \tag{18}$$

**Total FLOPs Overhead:** The total computational overhead ratio is:

$$R_{\text{FLOPs}} = \frac{\Delta C_{\text{anchor}} + \Delta C_{\text{DM}}}{C_{\text{base}}} = \frac{8d^2 + L(2dd_{\text{DM}} + 12d)}{22Ld^2} = \frac{8}{22L} + \frac{d_{\text{DM}}}{11d} + \frac{6}{11d}. \tag{19}$$

For values ($L = 32, d = 1024, d_{\text{DM}} = 16$), the overhead is dominated by the anchor projection ($\approx 1.1\%$), followed by the dynamic module projection ($\approx 0.14\%$), and the elementwise mixing operations ($\approx 0.05\%$), resulting in a total FLOPs increase of approximately $1.33\%$.

### F.3 EFFICIENCY-PERFORMANCE TRADE-OFF

The FLOPs analysis presented above provides a lower-bound estimate of the computational overhead. In practice, the real-world slowdown may be higher (we observe an increase in latency per token of approximately 8–15% for our reference implementation prioritizing modularity and interpretability over speed, compared to the theoretical 1.3% FLOPs increase).

We emphasize that our primary contribution is a novel architecture that unifies attention projection mixing, not a production-optimized layer ready for deployment. The net performance benefit even in our setting, evidenced by accuracy gains of 1.5 points and data efficiency improvements of 1.5× against Gated Attention, remains positive even when accounting for the measured runtime overhead, suggesting that an optimized kernel implementation would close the gap with the theoretical 1.3% overhead.

## G    HYPERPARAMETERS

Table 3: Training hyperparameters for models with different configurations for the main $\sim 450M$ models. Layer depth was reduced for gated variants to maintain comparable parameter counts. All models were trained on the FineWeb-Edu dataset.

| Hyperparameter | No Gate | With Gate | Decoupled (ExoFormer) |
|---|---|---|---|
| Parameters (M) | 454 | 453 | 457 |
| Layers | 32 | 29 | 29 |
| Attention Heads | | 16 | |
| Hidden Dimension | | 1024 | |
| FFN Dimension | | 4096 | |
| Tie Word Embedding | | False | |
| Vocabulary Size | | 57,601 | |
| Activation Function | | SwiGLU | |
| Position Embedding | | RoPE ($\theta = 500{,}000$) | |
| Sequence Length | | 2048 | |
| Batch Size (tokens) | | 262,144 | |
| Training Tokens | | 10B | |
| Warmup Steps | | 1000 | |
| Warmdown Steps | | 7630 (20%) | |
| Total Steps | | 38,147 | |
| **Optimization** | | | |
| Optimizer | | Muon + AdamW | |
| Muon Learning Rate | | 0.01 | |
| AdamW Learning Rate | | 0.003 | |
| Learning Rate Schedule | | Linear | |
| Adam $\beta$ | | (0.9, 0.95) | |
| Muon Momentum | | 0.95 | |
| Gradient Clip | | 1.0 | |
| Dropout | | 0.0 | |
| Cautious Weight Decay | | True | |
| Muon Weight Decay | | 0.1 | |
| AdamW Weight Decay | | 0.0 | |
| Z Loss Weight | | 1e-5 | |
| RMSNorm Epsilon | | 1e-6 | |
| QK Normalization | | True | |

Table 4: Training hyperparameters for 1B parameter models. Most hyperparameters remain identical to those in Table 3 except for depth, width, and learning rates. Batch size was kept constant, requiring an increase in total steps to reach 20B training tokens.

| Hyperparameter | Gated Attention | Dynamic ExoFormer |
|---|---|---|
| Parameters (B) | 1.01 | 1.02 |
| Layers | | 32 | |
| Hidden Dimension | | 1536 | |
| FFN Dimension | | 6144 | |
| **Optimization** | | |
| Muon Learning Rate | | 0.003 | |
| AdamW Learning Rate | | 0.001 | |
| Training Tokens | | 20B | |
| Total Steps | | 76,293 | |

## H  SCIENCE OF DL IMPROVEMENT CHALLENGE SUBMISSION

### H.1  WHAT MODEL ARE YOU TARGETING?

*Provide a summary of the problem the deep net model is designed to solve. Good summaries should outline the state of the literature, provide an overview that domain experts would consider reasonable, and cite relevant sources.*

We target the Transformer architecture (Vaswani et al., 2017), specifically its application in Large Language Models (LLMs). The core problem addressed is the degradation of information propagation and optimization efficiency as model depth increases, manifested as over-smoothing (Shi et al., 2022; Zhou et al., 2021) and the inefficient computational cycle of "Mix-Compress-Refine" (Queipo-de-Llano et al., 2025). While existing work like ResFormer (Zhou et al., 2025) and Gated Attention (Qiu et al., 2025) mitigate these issues by residualizing value projections or modulating attention, they operate in isolation and a naïve combination of both under-delivers in performance.

### H.2  HOW DO YOUR RESULTS CONTRIBUTE—OR COULD POTENTIALLY CONTRIBUTE—TO UNDERSTANDING THESE MODELS?

*What aspects of the models become better understood thanks to your work?*

Our work advances the understanding of Transformer dynamics in three ways: (1) We empirically link the performance gains of Gated Attention and ResFormer to their effect on the "compression" stage of the Mix-Compress-Refine cycle. We show that these mechanisms reduce the model's reliance on drastic attention-sink-based compression, thereby reducing inefficiency. (2) Through gating profile analysis, we reveal that internal-anchor designs (NuResFormer) force the first layer into a suboptimal compromise, adopting a permissive gating policy to serve as a usable anchor, sacrificing its ability to select and transform features effectively. (3) We propose and validate the hypothesis that decoupling identity preservation (via exogenous anchors) allows sequential layers to specialize exclusively in the "refinement" stage. We demonstrate that ExoFormer layers spend approximately two-thirds of their depth in refinement (compared to one-third in standard models) and suffer catastrophic failure upon anchor removal, proving they rely on the external anchor to maintain token identity.

### H.3  HOW DO YOU EXPECT YOUR SUBMISSION TO INFLUENCE FUTURE WORK?

*Propose ways in which your insights, findings, or methodologies could shape subsequent research directions, model design choices, or scientific applications.*

We anticipate that our findings will encourage future research to explore more ways for architectural decoupling in deep stacks. Our findings suggest that architectural decoupling (separating identity preservation from feature transformation) may offer a viable path toward mitigating limitations inherent in standard residual mixing. Furthermore, our unified normalized mixing framework enables reliable cross-layer reuse of queries and keys, resolving instability that has hindered prior approaches.

