# OpenReview forum: "Attention Projection Mixing with Exogenous Anchors"
_ICLR.cc/2026/Workshop/Sci4DL — Sci4DL 2026_

### Official Review · Reviewer_VdPU · 2026-02-27

**Fit:** 2
**Significance:** 2
**Confidence:** 2

**Summary:**

This paper studies cross-layer reuse of attention projections and argues that “internal-anchor” designs (reusing layer-1 projections as anchors) face a structural conflict: the first layer must be both a strong compute block and a stable, reusable reference. The authors propose ExoFormer, which instead learns exogenous anchor projections directly from input embeddings (outside the layer stack) and mixes anchors into each layer’s Q/K/V and gate logits with learnable coefficients. A key stabilizer is RMSNorm on the anchor source before mixing. Experiments at ~450M and ~1B params on FineWeb-Edu show ExoFormer (especially a dynamic mixing variant) improves perplexity and yields ~1–1.5 average downstream accuracy points over a gated-attention baseline.

**Strengths:**

This work proposed a new architecture that follows recent trends to consider cross-layer connectivity methods with a valuable hypothesis (Offloading hypothesis) that is helpful to design frontier architectures, and should be accepted given the context of this workshop avenue.
- The “first-layer as anchor” tension is intuitive, and using exogenous anchors cleanly decouples the anchor role from the layer stack (NuResFormer vs ExoFormer).
- Mixing $Q/K/V/G$ under a single normalized framework goes beyond value-only reuse, and the granularity sweep (scalar/headwise/elementwise) is practically informative.
- The results suggest anchor-source normalization reduces distribution mismatch and prevents the model from effectively disabling the anchor path.

**Suggestions:**

**Clarity**
- The authors should clarify where exactly RoPE and QKNorm are applied relative to mixing, and why normalize only the anchor source, not both anchor and current projections.

**Evaluation**:
- I recommend that the author should compare ExoFormer with recent cross-layer connectivity methods: Hyper-Connections, mHC, ... if the authors consider submitting this work to a flagship conference. The lack of comparison with these baselines is the main reason why I cannot raise the higher score.
- A table about throughput, memory to reconcile FLOPs vs real latency would be helpful for the community

---

### Official Review · Reviewer_Umyq · 2026-03-01

**Fit:** 2
**Significance:** 2
**Confidence:** 2

**Summary:**

This paper proposes ExoFormer, a Transformer variant that improves cross-layer information reuse by introducing exogenous anchor projections. These are attention projections computed outside the sequential layer stack and mixed into each layer’s Q/K/V/G projections using normalized learnable coefficients. The authors argue that prior internal-anchor approaches create a structural tension: the first layer must both preserve stable token identity for reuse and perform useful feature transformations, limiting both roles. They formalize a unified mixing framework in which each projection is replaced with a normalized combination of current-layer signals and anchor signals.
They show that RMS normalization of anchors stabilizes training by resolving distributional mismatch. Experiments on indicate that ExoFormer variants consistently outperform internal-anchor counterparts and can achieve about $1.5\times$ data efficiency relative to Gated Attention. The authors interpret results through an Offloading Hypothesis: external anchors preserve token identity so stacked layers specialize in refinement rather than reconstruction, supported by representation analyses showing longer refinement phases and catastrophic collapse when anchors are removed.

**Strengths:**

1. The paper identifies a clear architectural tension in cross-layer reuse and proposes a simple structural fix.

2. The empirical study is thorough in ablations: normalization variants, coefficient granularities, internal vs external anchors, dynamic vs static mixing, and comparisons against multiple baselines. The parameter and FLOPs analyses show the method is not just effective but also efficient.

3. Interpretability analysis is stronger than typical architecture-focused papers. The representation-similarity plots, entropy curves, and gating-profile diagnostics provide some mechanistic evidence consistent with the proposed hypothesis.

**Suggestions:**

1. The main theoretical claim (the Offloading Hypothesis) is supported primarily by qualitative representation diagnostics and ablations. A more formal analysis would strengthen the causal argument that external anchors enable specialization rather than simply correlate with it.

2. The evaluation focuses on mid-scale models ($\leq 1$B parameters). Because the motivation concerns deep-network scaling and information dilution, validation on substantially deeper or larger models would be important to verify that the benefit persists in regimes where residual pathways already mitigate signal loss.

3. The comparison against other cross-layer communication methods (e.g., hyper-connections or dense skip variants) is mostly conceptual; direct empirical comparisons would clarify whether the gains stem from exogenous anchoring specifically or from increased cross-layer signal bandwidth more generally.

---

### Official Review · Reviewer_PQPq · 2026-03-02

**Fit:** 2
**Significance:** 2
**Confidence:** 2

**Summary:**

The paper claims to identify a "structural conflict" in Transformers where early layers are burdened with being both identity anchors and functional blocks. The authors propose a new architecture ExoFormer, to decouple these roles and allow deeper layers to focus purely on feature refinement

**Strengths:**

The paper attempts to scientifically separate "identity preservation" from "feature refinement," which is an interesting conceptual lens for looking at Transformer depth.

**Suggestions:**

I understand this is a work in progress and that this is a workshop submission, but I don't feel the problem being studied is well motivated. The "structural conflict" described isn't clearly proven to be a bottleneck that standard residual connections can't already handle. The paper fails to convincingly demonstrate why moving anchors outside the stack is a necessary or superior solution. Moreover, the results don't seem super convincing, and it might need more empirical validation.

---

### Meta-Review · Area_Chair_JMuc · 2026-03-01

**Recommendation:** Accept

**Metareview:**

The main theoretical claim is supported through experimental analysis, and the findings are worth sharing with the community in the workshop. I recommend acceptance.

---

### Decision · Program_Chairs · 2026-03-02

Accept